# Effects of Antioxidant Gene Overexpression on Stress Resistance and Malignization In Vitro and In Vivo: A Review

**DOI:** 10.3390/antiox11122316

**Published:** 2022-11-23

**Authors:** Marina M. Tavleeva, Elena S. Belykh, Anna V. Rybak, Elena E. Rasova, Aleksey A. Chernykh, Zaur B. Ismailov, Ilya O. Velegzhaninov

**Affiliations:** 1Institute of Biology of Komi Scientific Centre, Ural Branch of Russian Academy of Sciences, 28b Kommunisticheskaya St., Syktyvkar 167982, Russia; 2Institute of Physiology of Komi Scientific Centre, Ural Branch of Russian Academy of Sciences, 50 Pervomaiskaya St., Syktyvkar 167982, Russia

**Keywords:** antioxidant genes, overexpression, stress resistance, carcinogenesis, oncosuppression, gene therapy

## Abstract

Reactive oxygen species (ROS) are normal products of a number of biochemical reactions and are important signaling molecules. However, at the same time, they are toxic to cells and have to be strictly regulated by their antioxidant systems. The etiology and pathogenesis of many diseases are associated with increased ROS levels, and many external stress factors directly or indirectly cause oxidative stress in cells. Within this context, the overexpression of genes encoding the proteins in antioxidant systems seems to have become a viable approach to decrease the oxidative stress caused by pathological conditions and to increase cellular stress resistance. However, such manipulations unavoidably lead to side effects, the most dangerous of which is an increased probability of healthy tissue malignization or increased tumor aggression. The aims of the present review were to collect and systematize the results of studies devoted to the effects resulting from the overexpression of antioxidant system genes on stress resistance and carcinogenesis in vitro and in vivo. In most cases, the overexpression of these genes was shown to increase cell and organism resistances to factors that induce oxidative and genotoxic stress but to also have different effects on cancer initiation and promotion. The last fact greatly limits perspectives of such manipulations in practice. The overexpression of *GPX3* and *SOD3* encoding secreted proteins seems to be the “safest” among the genes that can increase cell resistance to oxidative stress. High efficiency and safety potential can also be found for *SOD2* overexpression in combinations with *GPX1* or *CAT* and for similar combinations that lead to no significant changes in H_2_O_2_ levels. Accumulation, systematization, and the integral analysis of data on antioxidant gene overexpression effects can help to develop approaches for practical uses in biomedical and agricultural areas. Additionally, a number of factors such as genetic and functional context, cell and tissue type, differences in the function of transcripts of one and the same gene, regulatory interactions, and additional functions should be taken into account.

## 1. Introduction

Reactive oxygen species (ROS) participate in the regulation of physiological processes, including gene expression regulation, the activation of stress-response signaling cascades, and apoptosis (for a review, see [1]). Mitochondria, which are the main source of ROS in the cell, use them as signal molecules [2]. At submicromolar concentrations, ROS induce the proliferative activity of cells, working as mitogens. It is known [3] that mitochondria can activate cellular immune response by changing their membrane potential and by directly interacting with Toll-like receptors as well as by participating in signaling that regulates innate immune responses. However, ROS can cause macromolecule damage [1], and therefore, their levels are strictly regulated by antioxidant systems [4].

Oxidative stress is a disbalance between ROS and reactive nitrogen species (RNS) on the one hand and antioxidant defense on the other. Among the causes of oxidative stress are the imperfect Red-Ox regulation of normal physiological processes, primarily anaerobic respiration, and oxidative phosphorylation in mitochondria [5], as well as decreased antioxidant protein levels due to pathological conditions or senescence. Oxidative stress can be the result of chemical and physical factors with direct ROS and RNS generation activity or can be achieved indirectly by acting through endocrine disruption [6,7,8]. Causality works both ways when it comes to the relationship between oxidative stress and inflammation. Through various signaling cascades, chronic oxidative stress can activate pro-inflammatory cytokines [9], and the inflammation itself, in turn, leads to the production of additional amounts of ROS [10]. The emerging vicious circle underlies many pathologies, including the initiation of carcinogenesis [9,11].

Oxidative stress is an important component in the etiology and pathogenesis of oncology [5,12], cardiovascular (ischemia-reperfusion) diseases [13], neurodegenerative diseases (Alzheimer’s, Parkinson’s diseases, etc.) [12,14,15], epilepsy [16], Huntington’s disease [17,18], diabetes mellitus [19], and multiple organ dysfunction syndrome [20]. In addition to that, oxidative stress is associated with mechanisms of psoriasis development [21,22,23,24] and with skin fibrosis in systemic scleroderma [25]. Oxidative stress, which is primarily caused by mitochondrial dysfunction, plays one of the leading roles in the processes of aging, both in tissues and in the whole organism [5,26,27]. All that makes our knowledge of the possibilities and consequences of cell antioxidant systems regulation very important for the development of strategies and approaches for reducing the negative effects of oxidative stress as well as for prophylaxis and the treatment of a wide variety of diseases. The main strategies can be classified into pharmacological, nutritiological, genetic, and ecological [28,29,30,31]. The latter is more concerned with preventing the formation of ROS where possible [28]. Naturally, the most straightforward strategies are built on pharmacological interventions, predominantly with various antioxidants. Studies on the topic include assessments of the effects of curcumin and curcuminoids [32,33], plant-derived polyphenols [34,35], flavonoids [36], boswellic acids [37], coenzyme Q10 [38], quercetin [39] N-acetylcysteine [40], monoclonal antibodies to cytokines (infliximab, tocilizumab [41]), and many other compounds and pharmaceuticals. The efficacy and success rates of pharmacological strategies are, as one can conclude, limited, with four possible reasons suggested: (1) vitamin antioxidants based upon in vitro studies may be inefficient scavengers in vivo; (2) detrimental ROS may be compartmentalized at sites within the cell that are inaccessible by the administered antioxidant; (3) antioxidants may possess toxicities that mask their advantageous effect; and (4) heterogeneity of the groups under study [42]. The use of nutritional approaches to treat and prevent diseases and to leveling unhealthy environment adverse effects comes from eating foods rich in antioxidants or food additives/drugs based on such foods [43,44]. However, systematic or meta-analysis results suggest that eating antioxidants is more often harmless, but there is insufficient evidence on the positive effects of this for disease treatment or for the improvement of life quality [43,45,46], or the benefits are only revealed under a strictly defined set of measures or under a strictly defined disease stage with the observance of dosages [47,48,49,50]. Genetic approaches involve the regulation of the cell’s own antioxidant system. The results obtained when discovering the targets for such regulation are reviewed below.

In addition, the rational control of antioxidant system activity is required for the regulation of organisms and for normal cells’ resistance to stressful and adverse conditions. It is also required to regulate the resistances of malignant cells and tissues to chemotherapeutics and irradiation. Previously, for example, gene therapies with constructs that transiently overexpress antioxidant genes were used to elucidate approaches for protecting normal tissues during tumor radiotherapy [51]. Taking into account that almost all intracellular systems, including antioxidant ones, consist of more than one protein, approaches utilizing transcription regulation of antioxidant genes in functionally rational combinations seem to be the most promising ones [52]. These approaches have become viable because of our vast knowledge of the functions of both individual-proteins and whole-protein subsystems and because of development of CRISPRa and CRISPRi technologies for the regulation of gene expression in cells. At the same time, one must take into account potential side effects. For example, the overexpression of a transcription factor gene *NFE2L2* (also known as *Nrf2*) can lead to the induction of skin anomalies in mice [53], and the overexpression of *SOD1* without the simultaneous overexpression of H_2_O_2_-utilizing enzymes genes lead to the faster loss of function in retina cones. *GPX4* overexpression decreased these negative effects of *SOD1* overexpression [54]. However, the most important finding was that the overexpression of particular antioxidant genes could increase or decrease the probability of malignization as well as differently affect processes in already-developed tumor tissues.

The aim of this review was to systematize the results of gain of function studies of antioxidant systems genes and to search for candidate genes and their combinations for the safe and effective treatment and prevention of oxidative stress with different genesis. Special attention was devoted to the results of experimental studies covering the effects resulting from the overexpression of particular antioxidant genes and their combinations on the stress resistance of cells, the probability of tumor induction, and tumor progression. We believe that systematization would be an important intermediate point for further studies for the manipulation and control of oxidative stress resistance in cells and organisms.

## 2. Loss of Function Studies of Antioxidant Genes

Obviously, knockout or knockdown genes, products of which are components of primary life-support systems ([55] for review), result in the decreased viability and stress resistance of cells and organisms and an increased probability of malignization in the majority of cases.

For example, the knockdown or knockout of antioxidant defense genes decreased cell resistance to pro-oxidants (*SOD2*: [56]; *SOD1*, *TRX1*: [57]; *PRXD2*: [58]; *PRDX1*: [59]; *CAT*: [60]; *PRDX2*: [61]), alkylating agents (*PRDX1*: [62]; *PRDX3*: [63]; *PRDX2*: [64]), and ionizing radiation (*PRDX1*: [65]; *PRDX4*: [66]; *PRDX1*, *PRDX5*: [67]; *GSL2*: [68]; *PRDX2*: [61,64]). At the scale of the whole organism, *PRDX1* knockdown in mice led to decreased lifespan, the development of hemolytic anemia, and malignant neoplasms (lymphomas, sarcomas, and carcinomas) [59]. The suppression of *NFE2L2* significantly affected the development of endothelial dysfunction and decreased microvascular bed density in normotensive rats under oxidative stress caused by NaCl [69]. Decreased *SOD1* activity elevated the neurotoxic potential of quinolonic and kainic acids in the corpus striatum of transgenic mice [70].

However, there are opposite examples of the effects of antioxidant gene knockout, knockdown, or suppression. *TRXR1* knockdown in A549 cells decreased their resistance to 1-chloro-2,4-dinitrobenzene or menadione, but at the same time, it significantly increased their resistance to cisplatin. This effect can be explained by TrxR1 protein derivatized with cisplatin being able to induce cell death [71].

The results of a number of loss of function studies were very important for deciphering the functions of many antioxidant genes and of the mechanisms of antioxidant systems. Presently, the elements of the reaction cascade that provides Red/Ox homeostasis are well-studied. These data are required for the development of new approaches to overcome tumor resistance to chemo- and radiotherapy as well as to decrease side effects in normal tissues and whole organisms [1,51]. However, much less is known about such effects on the cellular level and at the organizational levels above it that are induced by the increased gene expression of antioxidant proteins, both individually and in various combinations. Although such interventions also interfere with a fine-tuned cell balance, the effects produced are significantly diverse, and not only bring new fundamental knowledge, but also have potential importance in applied contexts. Therefore, the systematic review we present here is focused on the available results of gain of function studies of antioxidant defense genes.

## 3. Effects of Antioxidant Defense Gene Overexpression on Stress Resistance

The main aim of this review was to systemize and sum up the available studies covering the effects of antioxidant defense gene overexpression on cell resistance to oxidative stress. We analyzed 166 such works [20,51,56,58,62,65,66,71,72,73,74,75,76,77,78,79,80,81,82,83,84,85,86,87,88,89,90,91,92,93,94,95,96,97,98,99,100,101,102,103,104,105,106,107,108,109,110,111,112,113,114,115,116,117,118,119,120,121,122,123,124,125,126,127,128,129,130,131,132,133,134,135,136,137,138,139,140,141,142,143,144,145,146,147,148,149,150,151,152,153,154,155,156,157,158,159,160,161,162,163,164,165,166,167,168,169,170,171,172,173,174,175,176,177,178,179,180,181,182,183,184,185,186,187,188,189,190,191,192,193,194,195,196,197,198,199,200,201,202,203,204,205,206,207,208,209,210,211,212,213,214,215,216,217,218,219,220,221,222,223,224,225,226,227] published during the period covering 1988–2022. Table 1 contains extremely condensed information on the effects of overexpression of the following antioxidant genes (with direct and indirect antioxidant action): *SOD1*, *SOD2*, *SOD3* (*EC-SOD*), *CAT*, *MTI*, *MTII*, *ALDH3A1*, *TRX1*, *TRX2*, *GCLC*, *GCLM*, *GPX1*, *TXN1*, *NQO1*, *PRDX1*, *PRDX2*, *PRDX3*, *PRDX4*, *PRDX5*, *PRDX6*, *TXD1*, *SLC31A1*, *SLC7A11*, *GPX2*, *GPX3*, *GPX4*, *GSTP1*, *SRXN1*, *TRXR1*, *PON1*, *PON2*, and *PON3*. More detailed information about the analyzed studies is presented in the Appendix A. Resistance in vivo or in vitro was assessed to determine the oxidative stress caused by various irradiation types (UV, X-rays, and gamma), chemical compounds (hydrogen peroxide, paraquat, etc.), and other factors (ischemia, sepsis, surgical interventions, hypoxia, and endotoxemia). In the majority of published studies that we analyzed, the overexpression of antioxidant defense genes in vitro and in vivo was achieved by introducing an additional copy of the gene or a transgene. CRISPRa technology, which became available in 2013 [228], despite its obvious advantages [52], was registered in one work [177].

Cases of spontaneous gene overexpression in tumors resistant to particular therapy types were also reviewed, but only in such cases where the abolition of overexpression of a particular gene led to the abolition of the changed phenotype.

In most of the studies analyzed, the overexpression of antioxidant defense genes resulted in increased resistance to oxidative stress induced by different factors, both in vitro and in vivo. For example, the increased survival of cells was found in NK-92 and NK-92MI exposed to H_2_O_2_ (*PRDX1*, [151]). Mice overexpressing *TRX1* survived better after adriamycin treatment [219], and mice overexpressing *PON1* or *PON3* had less liver damage after treatment with carbon tetrachloride [135]. Decreased apoptosis levels were observed for *PRDX6* in vitro in SCOV-3 cell lines treated with cisplatin [169], for *PRDX5* in human tendon fibroblasts treated with H_2_O_2_ [161], and in HT29 cells treated with shikonin [166], for *TRX1* in WEHI7.2 lymphoma cells treated with H_2_O_2_ [220], in in vivo studies on transgenic mice that were treated with methamphetamine [214], and for *SRXN1* in retinal ganglion cells (RGC) of mice treated with high glucose concentrations [229]. Among the positive effects of overexpression, for *PRDX5*, researchers have shown decreased DNA damage in hamster cell cultures CHO-K1 exposed to H_2_O_2_ and tert-butylhydroperoxide (tBHP) [162] as well as for *TRX1* in the mitochondria of transgenic mice Trx1-Tg in vivo in sepsis-induced myocardial dysfunction and sham surgery [216,217]. Increased expression of *TRX1* in mice in vivo [217] and ex vivo [218] decreased the negative consequences of ischemia for the cardiovascular system. Using oxygen–glucose deprivation/reoxygenation in in vitro experiments showed that the ability of *PON2* to enhance glucose transport and to suppress oxidative stress and apoptosis is potentially a good set of properties to prevent cerebral ischemia-reperfusion injury [143].

In some cases, the authors noted that increased expression of the antioxidant defense genes in fact decreases cell resistance to oxidative stress; however, such studies are scarce. For example, decreased resistance to oxidative stress was shown in the brain neurons of mice overexpressing *SOD1* when exposed to menadione [182], in human cells overexpressing *NFE2L2* that were treated with cisplatin [126,129], and in human cells overexpressing *GPX1* when exposed to radiation [100]. No changes in stress resistance were found in human cells TK6 overexpressins *SOD1* when treated with gamma-irradiation [181]; in normal human keratinocytes overexpressing *SOD1* when irradiated with UV [73]; in brain neurons of mice overexpressing *SOD1* treated with H_2_O_2_ [182]; in Chinese hamster ovary cells overexpressing *MTII* when treated with gamma-rays, bleomycin, MMS, and N-hydroxyethyl-N-chloroethylnitrosourea [122]; in mouse C127 cells overexpressing *MTII* that had been exposed to 5-fluorouracil and vincristine [123]; in Chinese hamster cells V79 overexpressing *MTI* when exposed to alkylating agents [121]; in vivo in *D. melanogaster* overexpressing *SOD1* when the flies were exposed to paraquat [185]; and in *SOD2*-overexpressing ones exposed to 100% O_2_ [199].

The number of studies that have shown the overexpression of various antioxidant defense genes increasing the oxidative stress resistance on the cell and organism levels gives ground to careful optimism when assessing the future feasibility of gene expression regulation in these genes for therapeutic purposes. However, the in vitro effects that are considered beneficial at that level might be dangerous or even harmful at higher levels of structural and functional organization. Similarly, relatively positive effects in in vivo models that have been observed to be beneficial at the endpoints selected by researchers might cause negative side effects in the long term according to other parameters. Taking this into account, an extra task was to assess the effects of the increased expression of the antioxidant defense genes on the probabilities of malignization in normal cells and on the growth and development of tumor cells.

## 4. Effects of the Antioxidant Defense Genes Overexpression on Carcinogenesis

Logically, the overexpression of antioxidant genes, as said above, has predominantly cytoprotective effects and must decrease the frequency of spontaneous and induced mutations and therefore must decrease the probability of malignization. On the other hand, in already malignantly transformed cells, it can stimulate faster proliferation, decrease apoptosis and permanent cell cycle arrest, and increase tumor resistance to therapeutic interventions. Previously, the authors of a study on *PRDX6* came to similar conclusions [230]. Another review devoted to glutathione peroxidases came to similar conclusions [231], with the authors stating that all *GPX*s prevent cancer initiation in normal cells by removing hydroperoxides, but that in already malignantly transformed cells, they have various effects, including tumor promotion. In a limited set of studies on *SOD3* overexpression, a completely opposite picture can be found. The increased expression of *SOD3* led to the immortalization of mouse embryonic fibroblasts and carcinogenesis despite the fact that, in the developed tumors, this gene’s expression is usually suppressed [232]. Additionally, there is direct evidence that *SOD3* overexpression suppresses growth and aggression in tumor cells both in vitro and in vivo [233,234]. Limited samples of focused studies of the overexpression of other genes also support and disprove the proposed simplified idea (see Figure 1 and Appendix A). Additionally, we often talk about one and the same gene. For instance, increased *SOD2* expression has often been observed in tumor tissues and is frequently correlated with tumor stage and metastasis activity [235,236]. More than that, some studies have demonstrated that *SOD2* overexpression in tumors promotes growth and metastasis [237,238] and increases resistance to anoikis [239]. However, there are many works that present evidence for the opposite. Cancer cells from the MCF-7, U118, and MIA PaCa-2 lines found to overexpress *SOD2* when injected into nude mice showed significantly lesser and slower tumor development and decreased mortality when compared with the same cell lines without *SOD2* overexpression [240,241,242]. An even more pronounced effect of decreased tumor growth and increased animal survival was achieved in MIA PaCa-2 cells when *SOD2* and *GPX1* were overexpressed simultaneously [243]. The overexpression of *SOD2* with the lentiviruses that deliver this gene with powerful promotors, in human cell cultures and in xenografts, and in surrounding tissues in mice simultaneously led to the decreased resistance and proliferation of cancer cells and provided protection against radiation damage for normal cells at the same time [206]. In other types of experiments, increased expression of *SOD2* also suppressed the development and metastasis of tumors. In contrast with the results of the study mentioned above, in which increased levels of H_2_O_2_ associated with *SOD2* overexpression stimulated the migration and invasion of HT-1080 cells [238], the same effects were achieved by suppressing *SOD2* expression in ovarian cancer cells resulting from increased superoxide–radical concentrations [244,245]. In other words, in different experimental systems, *SOD2* overexpression can either stimulate tumor growth and metastasis or suppress them. It seems that an important role belongs to the ratios of expression of superoxide dismutases with expression of *CAT* and/or *GPX*s, the protein products of which have hydrogen peroxide as a substrate [236]. As a consequence, H_2_O_2_ and superoxide radicals probably are among the key signal molecules involved in the regulation of such processes [238,244,245,246,247,248].

The avoidance of H_2_O_2_ overproduction is likely to be the cause of no promotion of cancer cells when there is simultaneous overexpression of *SOD2* and *CAT* both in vitro [247] and in vivo [246] and of tumor-preventive [249] and tumor-suppressive [243] effects in the case of the simultaneous overexpression of *SOD2* and *GPX1*. This is another argument for higher efficacy antioxidant gene overexpression in functionally viable combinations in contrast with the overexpression of individual genes.

A graphical representation of the effects of the overexpression of antioxidant genes on the induction, prevention, promotion, or suppression of tumor processes (based on an analysis of 130 studies) [58,63,66,87,88,89,90,103,104,105,114,117,159,173,174,175,192,201,210,230,232,233,234,238,239,240,241,242,243,244,245,246,247,248,249,250,251,252,253,254,255,256,257,258,259,260,261,262,263,264,265,266,267,268,269,270,271,272,273,274,275,276,277,278,279,280,281,282,283,284,285,286,287,288,289,290,291,292,293,294,295,296,297,298,299,300,301,302,303,304,305,306,307,308,309,310,311,312,313,314,315,316,317,318,319,320,321,322,323,324,325,326,327,328,329,330,331,332,333,334,335,336,337,338,339,340,341,342,343] is given in Figure 1. More detailed information and references to actual studies are provided in Appendix A.

## 5. Problems and Perspectives of Applied Regulation of Cell Stress Resistance by Overexpressing the Antioxidant Defense Genes

Based on the above, the overexpression of antioxidant defense genes has practical potential for the therapeutic regulation of oxidative stress and for increasing cellular stress resistance. On the other hand, studies on side effects of such overexpression, primarily its cancerogenic potential, give very contradictory results, and we have to first elucidate which genes and in which context overexpression can occur without incurring unjustified risks.

For example, *SOD2* overexpression, the consequences of which are apparently the best-studied ones, seems to be counterproductive without the simultaneous overexpression of genes, products of which eliminate H_2_O_2_, such as *CAT* or *GPXs*. Among the last group the most applicable ones are *GPX1* [243,249], the overexpression of which by itself has different effects on cancer initiation and development in different experimental systems (cancer initiation [249]; promotion [103,105,301]; and suppression [104,243]). The same is true for *GPX4* [231,340]. Catalase looks like an obvious and even more viable partner for the simultaneous overexpression with *SOD2*. Firstly, for this gene combination, we already know the effect of increased cell resistance to gamma rays [82]. Secondly, it was reported that catalase prevents *SOD2* overexpression-induced promotion of cancer cells in vitro [247] and in vivo [246]. Thirdly, the isolated overexpression of *CAT* in experiments led to the prevention of malignant transformation [87,318,319,320,321] or to the suppression of tumor cells growth [89,90] significantly more frequently than it did to cancer promotion [88]. It is possible that cytoplasmic superoxide dismutase (*SOD1*) is an even more “suitable” partner for *CAT* overexpression because for this combination, increased cell resistance to UV [73], paraquat, and H_2_O_2_ [83] were observed in vitro, and increased cell resistance to benzopyrene was observed in vivo [75]. However, data on effects of the simultaneous expression of *CAT* and *SOD1* on cancer promotion are not available.

A special attention is given to one other member of the glutathione peroxidase family s*GPX3*. Its overexpression increased cell resistance to ascorbate [109] and cisplatin [110]. Despite the fact that these findings were made in tumor cells that spontaneously overexpressed *GPX3* and that high *GPX3* expression was associated with poorer overall survival in patients with ovarian cancer and with increased tumor stage [109], all direct manipulations with *GPX3* overexpression conversely indicate the potential safety of such manipulation from the perspective of carcinogenesis. The suppression of proliferation, migration, and invasion in vitro [326,327,332,333] as well as of tumorigenicity in vivo [328,329,330,331] were shown in models overexpressing *GPX3*. Additionally, several other studies have provided evidence that in tumor tissues and cancer cell lines, *GPX3* expression is suppressed or completely blocked due to promoter hypermethylation or gene deletion [329,331,344,345,346]. Thus, GPX3 functions as a tumor suppressor in an overwhelming majority of studies, and its overexpression, isolated or in combination with other components of the antioxidant system, is very promising for the suppression of endogenous or exogenous oxidative stress.

*GPX3* encodes a secreted protein; therefore, *SOD3*, which encodes extracellular superoxide dismutase, could be a potentially effective functional partner for it. We know that transgenic mice that overexpress *SOD3* have increased resistance to rib cage irradiation with 4-MV photons [208], to focal cerebral ischemia [209], to 12-O-tetradecanoylphorbol-13-acetate (TPA) exposure [210], and to lipopolysaccharide-induced endotoxemia [20]. At the same time, *SOD3* overexpression decreases dimethylbenzanthracene/TPA-induced tumor formation [210] and suppresses cancer cells’ growth in vitro [233,234] and in vivo [234]. However, increased expression of *SOD3* in mouse embryonic fibroblasts, can induce carcinogenesis [232].

All examples of genes and gene combinations that have been discussed above are by no means an exhaustive list of potentially effective and safe targets for therapeutic overexpression. Seeming contradictions in the effects resulting from the overexpression of some antioxidant genes on stress resistance and carcinogenesis were, apparently, caused by a number of non-obvious factors. In some cases, different effects of overexpression of the same gene could have risen from functional differences between the isoforms of product protein created by alternative splicing of the same gene. For example, the activity of invasion and metastasis of colon cancer cells increased when a splice variant b of the *GLRX3* gene (Txl-2b) was expressed, decreased when *GLRX3c* was expressed, and did not change during *GLRX3a* expression [294]. The activities of different isoforms of glutaminase, which is involved in the glutathione-dependent antioxidant defense system of the cell, could either promote tumor progression or suppress it through certain pathways involving miRNAs [347]. Besides different isoforms of proteins transcribed from one gene, non-coding circular RNAs (circRNAs) that have regulatory functions can be transcribed. For example, circRNA transcribed from the *SOD2* gene plays an important role in hepatocellular carcinoma progression [348]. Therefore, different functional properties of different transcripts of the gene must be taken into account when choosing cDNA for the overexpression of the gene by a classic method via the introduction of an additional copy of the gene or when designing guide RNAs for promoters of genes that have several alternative transcription initiation sites (such as *SOD2*) for CRISPRa technology application. However, these features of genes are discussed far less than the effects of the simultaneous overexpression of functional gene combinations are.

Nevertheless, the selection of only one splice variant of the gene is not supposed to be a perfect decision. Dysregulation of the gene’s expression because of the introduction of cDNA under a synthetic promoter or another gene’s promoter can cause contradicting effects, as, obviously, the degree of overexpression, its changes with time, and the presence or absence of a transcriptional response to various signals would affect the phenotypic manifestation of the gene. Preserving the chromosome context and, to some degree, the natural regulation when overexpressing a cell’s own genes with technologies, such as CRISPRa, could provide better results. In addition to that, CRISPRa allows the simultaneous overexpression of combinations of a number of genes. Presently, there are very few studies in which the researchers have overexpressed a cells’ own antioxidant genes.

Even the stage of tumor development may determine contradictions in the effects of antioxidant defense gene overexpression, apart from those related to the functional properties of different transcripts of the same gene and such obvious factors as the genetic context and the tissue (if it is a local gene therapy). Thus, *SOD2* overexpression in mice with benign thyroid tumors resulted in an increased tumor burden. In contrast, in mice with aggressive follicular thyroid cancer, the overexpression of *SOD2* reduced tumor proliferation and improved mortality rates [259].

Knowledge of the cytoprotective potential of even those genes whose overexpression has a pronounced pro-oncogenic effect can be of high applied value. For example, the overexpression of *PON2* in most experimental systems led to cancer promotion [313,314,315,316]. The overexpression of *PON2* is found in many tumors and often correlates with tumor aggressiveness and poor prognosis [315,316,349]. At the same time, a unique set of properties of the product of this gene, including the ability to activate glucose transport [314,316] and to suppress oxidative stress and apoptosis [350], makes its temporary or local activation a promising manipulation to reduce the damage caused by ischemia-reperfusion [143].

## 6. Conclusions

The complexity and ambiguity of the effects of antioxidant defense gene overexpression also resulted from the multifunctionality of product proteins. In addition to their main function of ROS and organic radical detoxification, they are frequent participants in signal cascades and are regulators of other protein activities [90,104,194], including those that occur due to changes in ROS levels as signaling molecules [1,2,6]. Thus, the overexpression of individual genes and their combinations that encode the effectors of the antioxidant system seems to be more promising than attempts to regulate the transcription factors affecting them, such as *NFE2L2*. Furthermore, common approaches based on weak effects on multiple functionally complementary targets, for example, when using plant extracts [351,352,353] in the therapy of various diseases, seem to be more practically applicable than those with strong effects on a single target.

## Figures and Tables

**Figure 1 antioxidants-11-02316-f001:**
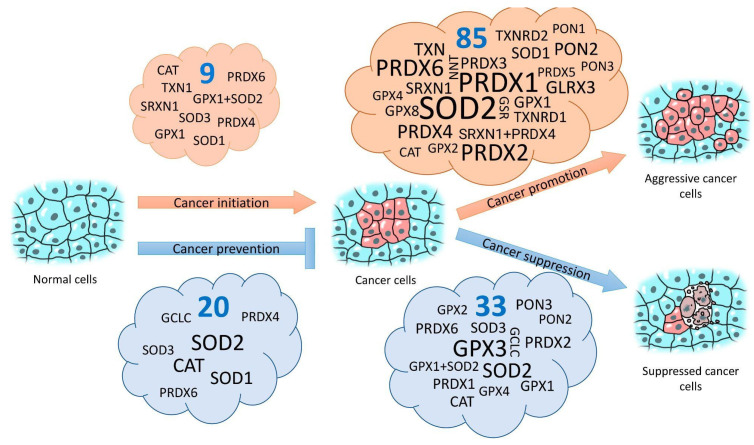
Effects of the overexpression of antioxidant defense genes on the probability of normal cells/tissues malignization and on the aggressiveness of already developed cancer cells/tumors. The font size is proportional to the number of studies that showed the effect of the overexpression of that gene. Numbers in blue show total numbers of studies in each group.

**Table 1 antioxidants-11-02316-t001:** Influence of antioxidant gene overexpression on resistance to oxidative and genotoxic stress factors in vitro and in vivo.

Gene	Model	Effect on Resistance
*ALDH3A1*	in vitro	Increased resistance: 4-hydroxyperoxycyclophosphamide, doxorubicin, etoposide, 5-FU, γ-ray, H_2_O_2_ [72]
*CAT*	in vitro	Increased resistance: UVB [73], benzo(a)pyrene [74], paraquat [81], H_2_O_2_ [83,89,90,91], benzyl isothiocyanate [84], dexamethasone [86], temozolomide, γ-ray, staurosporine [88], combination of ascorbate and menadione [90] Unchanged resistance: γ-ray [82], paraquat, TBHP [83], TNF-alpha [85], γ-ray, 5-FU, cisplatin, doxorubicin [90] Decreased resistance: Cr (VI) [87], paclitaxel, etoposide, arsenic trioxide, aminotriazol [90]
	in vivo	Increased resistance: benzo(a)pyrene [75,76], H_2_O_2_ [77], doxorubicin [78], proton radiation [79,80] Unchanged resistance: γ-ray [82]
*SOD1 +* *CAT*	in vivo	Increased resistance: benzo(a)pyren [75]
*GCLC*	in vitro	Increased resistance: γ-ray [93], UVC [94], tamoxifen [95]
	in vivo	Increased resistance: paraquat [92]
*GCLC+* *GCLM*	in vitro	Increased resistance: γ-ray [93]
*GCLM*	in vitro	Increased resistance: γ-ray [93]
*GPX1*	in vitro	Increased resistance: UVB [96], deoxynivalenol [97], cisplatin [99,103,105], X-ray [100], doxorubicin [106], H_2_O_2_, GSH high concentration [107] Decreased resistance: cisplatin [103], gemcitabine [104]
	in vivo	Increased resistance: ischemia-reperfusion injury [98], cocaine [101], microcystin-leucine-arginine [102]
*GPX2*	in vitro	Increased resistance: cisplatin [108]
*GPX3*	in vitro	Increased resistance: ascorbate [109], cisplatin [110], NAPQI [111]
*GPX4*	in vitro	Increased resistance: hypericin + PDT [112], TBHP [113], H_2_O_2_, erastin [114], AAPH, 5-HPETE [115], staurosporine, etoposide, UVB, actinomycin D, cycloheximide [116] Unchanged resistance: etoposide [113], calcium ionophore A23187 [116]
*GSR*	in vitro	Increased resistance: temozolomide, cisplatin [117], H_2_O_2_ [118]
*GSTP1*	in vitro	Increased resistance: hypericin+PDT [112]
*mt-PRDX5*	in vitro	Increased resistance: H_2_O_2_ [119]
*mt-SOD1*	in vitro	Increased resistance: γ-ray [51]
*mt-CAT*	in vitro	Increased resistance: γ-ray [82]
*mt-CAT + SOD2*	in vitro	Increased resistance: γ-ray [82]
*MTI*	in vitro	Increased resistance: TBHP, CdCl_2_ [120], Zn(II) [121] Unchanged resistance: alkylating agents [121]
*MTII*	in vitro	Increased resistance: MNU, MNNG [122], cisplatin, melphalan, chlorambucil [123], streptozotocin [124] Unchanged resistance: bleomycin [122,123], γ-ray, MMS, N-hydroxyethyl N-chloroethylnitrosourea [122], 5-FU, vincristine, doxorubicin [123]
*NFE2L2*	in vitro	Increased resistance: UVA+UVB [125], cisplatin [126,128,129,130], 5-FU [126,130], γ-ray [126], TBHP [127], X-ray [73], tamoxifen [131], gemcitabine [132]
*NQO1*	in vitro	Increased resistance: tamoxifen [95]
*nuclear-PRDX5*	in vitro	Unchanged resistance: H_2_O_2_ [119]
PON1	in vitro	Increased resistance: 5-fluorouracil, paclitaxel, cisplatin, etoposide [133], streptozotocin (mouse model of diabetes) [138]
	in vivo	Increased resistance: arthritis induction (K/BxN serum transfer (STIA) or collagen antibody transfer (CAIA) to mice [134], carbon tetrachloride [135,137], elastase-induced abdominal aortic aneurysm [136]
PON2	in vitro	Increased resistance: y-ray [139], tert-butyl-hydroperoxide [140], cisplatin [141], dexamethasone [142], oxygen-glucose deprivation/reoxygenation exposure [143], Nε-(carboxymethyl) lysine [144], pyocyanin [145], hypoxia [146], 2,3-dimethoxy-1,4-naphthoquinone [148] Unchanged resistance: dacarbazine [141]
	in vivo	Increased resistance: dexamethasone [142], apolipoprotein E deficiency [147]
PON3	in vivo	Increased resistance: carbon tetrachloride [135,149]
*PRDX1*	in vitro	Increased resistance: thiosemicarbazone iron chelator triapine [150], glucose oxidase [151], BCNU [62], γ-ray [65,152], 6-hydroxydopamine [152], doxorubicin [154]
*PRDX2*	in vitro	Increased resistance: H_2_O_2_ [58,155], γ-ray [156], 6-hydroxydopamine [153], doxorubicin [154]
	in vivo	Increased resistance: 6-hydroxydopamine [153], ischemia [176]
*PRDX3*	in vitro	Increased resistance: H_2_O_2_ [157], doxorubicin [154], arsenic trioxide [158] Unchanged resistance: 6-hydroxydopamine [153]
*PRDX4*	in vitro	Increased resistance: γ-ray [66], urethane [159], photon radiation [160], 6-hydroxydopamine [153] Unchanged resistance: doxorubicin [154]
*PRDX5*	in vitro	Increased resistance: H_2_O_2_ [161,162,165], TBHP [162], MPP+ [163], adriamycin, etoposide [164], menadione, cigarette smoke extract [165], shikonin [166], emodin [167], β-Lapachone [168], doxorubicin [154]
*PRDX6*	in vitro	Increased resistance: cisplatin [169], doxorubicin [170], hypoxia, cobalt chloride [171], UVA, menadione [172], H_2_O_2_ [173], X-ray [175] Decreased resistance: TNF-α/CHX solution [173]
	in vivo	Increased resistance: UVA, UVB [172], urethane [174]
*SLC31A1*	in vitro	Increased resistance: paraquat [177]
*SLC7A11*	in vitro	Increased resistance: cisplatin [178], temozolomide [179]
*SOD1*	in vitro	Increased resistance: γ-ray [181,183], S-nitroso-N-acetylpenicillamine, spermine-NONOate, diethylamine-NONOate [182], paraquat [83,177], cisplatin [186,187], docosahexaenoic acid [188], xanthine oxidase, hypoxanthine, menadione [189], TBHP [83], benzyl isothiocyanate [84], TNF and hyperthermia [190], Bortezomib [191], phenazine methosulfate [192] Unchanged resistance: γ-ray [51,180,193], N-methyl-D-aspartate, kainic acid, glutamate [182], H_2_O_2_ [83,182], UVB [73] Decreased resistance: menadione [182], Cr (VI) [87]
	in vivo	Increased resistance: paraquat, γ-ray [184], benzo(a)pyrene [75,76] Unchanged resistance: hyperoxia, paraquat [185]
*SOD1 + CAT*	in vitro	Increased resistance: paraquat, H_2_O_2_ [83], UVB [73] Unchanged resistance: TBHP [83]
*SOD2*	in vitro	Increased resistance: paraquat [180], γ-ray [51,82,181,195,196,203,205,206], cisplatin [194,201], X-ray [197], doxorubicin [198,200], docetaxel [200], 2-methoxyestradiol [202], glutamate [56], TNF- alpha [85], H_2_O_2_ [204], menadione, epirubicin [207] Unchanged resistance: γ-ray [193,206] Decreased resistance: Cr (VI) [87]
	in vivo	Increased resistance: γ-ray [51,82] Unchanged resistance: 100% O_2_ [199]
mt-signal deleted *SOD2*	in vitro	Unchanged resistance: γ-ray [51]
*SOD2 + CAT*	in vitro	Unchanged resistance: TNF-alpha [85]
*SOD3*	in vitro	Increased resistance: H_2_O_2_ [211]
	in vivo	Increased resistance: 4-MV photon radiation [208], ischemia [209], 12-O-tetradecanoylphorbol-13-acetate [210], lipopolysaccharide induced endotoxemia [20]
*TXN*	in vitro	Increased resistance: H_2_O_2_ [212,220,224], daunomycin [213], MPP+ [215], menadione, 1-chloro-2,4-dinitrobenzene [71], arsenic trioxide [222,223] Unchanged resistance: dexamethasone, doxorubicin, etoposide [220], auranofin, juglone [71]
	in vivo	Increased resistance: methamphetamine [214,221], sepsis-induced myocardial dysfunction, sham surgery [216], ischemia [217,225], ex vivo ischemia [218], adriamycin [219]
*TXNL1*	in vitro	Decreased resistance: cisplatin [226]
*TXNRD1*	in vitro	Increased resistance: X-ray [227], H_2_O_2_ [224]

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
