# Peer review of "Effects of Antioxidant Gene Overexpression on Stress Resistance and Malignization In Vitro and In Vivo: A Review"

_antioxidants, 2022, doi:10.3390/antiox11122316_

Round 1

Reviewer 1 Report

The manuscript “Effects of antioxidant genes overexpression on stress resistance and malignization in vitro and in vivo: a review” is a review article regarding the state-of-art of the studies about the effects of antioxidant system genes overexpression on stress resistance and cancerogenesis. There are several important concerns about this manuscript:

1.       The manuscript requires a language revision by a native English speaker.

2.       The manuscript seems to have been written carelessly. In line 47 there is the expression “it’s” which should be avoided; In line 299 there is a “(“ which never closes itself, etc..

3.       The major concern is that this manuscript completely ignored some antioxidant enzymes whose loss of function studies has been performed after assessing their overexpression in several tumors. For instance, Paraoxonase-2 (PON2) enzyme has been reported to be overexpressed in several tumors and its in vitro downregulation was proved to sensitize cells to chemotherapeutics in many models (melanoma, bladder cancer...)

4.       The table 1 is confused and not easy to be read.

5.       Figure 1 is not professional. Please revise it accordingly.

Author Response

The manuscript “Effects of antioxidant genes overexpression on stress resistance and malignization in vitro and in vivo: a review” is a review article regarding the state-of-art of the studies about the effects of antioxidant system genes overexpression on stress resistance and cancerogenesis. There are several important concerns about this manuscript:

Dear Reviewer. Thank you for your very valuable comments, which was helpful to improve the manuscript.

  1. The manuscript requires a language revision by a native English speaker.

The revised version of the manuscript was edited by MDPI language editing service.

  1. The manuscript seems to have been written carelessly. In line 47 there is the expression “it’s” which should be avoided; In line 299 there is a “(“ which never closes itself, etc..

Unfortunately, we did not see several inaccuracies. In the revised version, we tried to fix all the shortcomings that we could see.

  1. The major concern is that this manuscript completely ignored some antioxidant enzymes whose loss of function studies has been performed after assessing their overexpression in several tumors. For instance, Paraoxonase-2 (PON2) enzyme has been reported to be overexpressed in several tumors and its in vitro downregulation was proved to sensitize cells to chemotherapeutics in many models (melanoma, bladder cancer...)

Special thanks for this comment. In the revised version, we have added information that we could find on the PON1, PON2 and PON3 genes to the entire structure of the review.

  1. The table 1 is confused and not easy to be read.

We tried to change the structure of the data presentation in the table and removed empty cells.

  1. Figure 1 is not professional. Please revise it accordingly.

We changed the "cells" in the figure, making it more rigorous.

Reviewer 2 Report

Manuscript titled "Effects of antioxidant genes overexpression on stress resistance and malignization in vitro and in vivo: a review" is a very interesting review describing the role of oxidative stress in cancer initiation, progression and survival. The overall structure is of good quality, methods and discussion are well organized and of good quality. Introduction should be improved in several parts, such as:

1. a proper descriprion of the endocrine disruptor exposure in humans that increase the oxidative stress and reduce the expression of antioxidant genes should be made. For example, bisphenol A and phtalates that are able to increase pro-inflammatory pathways in human cardiac cells and breast cancer cells ( cite 10.1016/j.etap.2019.03.006)

2. Authors should add a paragraph that describe the pharmacological and non pharmacological strategies that reduces oxidative stress in clinical trials. For example, curcuminoids ( in proper formulations), boswellic acid, small chain fatty acid and other that are able to reduce lipid peroxidation, call apoptosis and necrosis induced by oxidative damages ( cite 10.1016/j.ijpharm.2015.08.039)

3. More data on cytokines that are involved in oxidative-related damages in human cancer cells ( interleukins and growth factors). 

Author Response

Manuscript titled "Effects of antioxidant genes overexpression on stress resistance and malignization in vitro and in vivo: a review" is a very interesting review describing the role of oxidative stress in cancer initiation, progression and survival. The overall structure is of good quality, methods and discussion are well organized and of good quality. Introduction should be improved in several parts, such as:

Dear Reviewer. Thank you very much for valuable comments. We agree that the information you provided is missing in the introduction. We have tried to add the essentials while keeping the presentation short, as we originally planned to write a small, informative review based on a large set of studies.

  1. a proper descriprion of the endocrine disruptor exposure in humans that increase the oxidative stress and reduce the expression of antioxidant genes should be made. For example, bisphenol A and phtalates that are able to increase pro-inflammatory pathways in human cardiac cells and breast cancer cells ( cite 10.1016/j.etap.2019.03.006)

A short snippet of text has been added: « Oxidative stress can be the result of chemical and physical factors with direct ROS and RNS generation activity or can be achieved indirectly by acting through endocrine disruption [6-8].»

  1. Authors should add a paragraph that describe the pharmacological and non pharmacological strategies that reduces oxidative stress in clinical trials. For example, curcuminoids ( in proper formulations), boswellic acid, small chain fatty acid and other that are able to reduce lipid peroxidation, call apoptosis and necrosis induced by oxidative damages ( cite 10.1016/j.ijpharm.2015.08.039)

The brief description of the main strategies with examples was added:

« All that makes our knowledge of the possibilities and consequences of cell antioxidant systems regulation very important for the development of strategies and approaches for reducing the negative effects of oxidative stress as well as for prophylaxis and the treatment of a wide variety of diseases. The main strategies can be classified into pharmacological, nutritiological, genetic, and ecological [28-31]. The latter is more concerned with preventing the formation of ROS where possible [28]. Naturally, the most straightforward strategies are built on pharmacological interventions, predominantly with various antioxidants. Studies on the topic include assessments of the effects of curcumin and curcuminoids [32,33], plant-derived polyphenols [34,35], flavonoids [36], boswellic acids [37], coenzyme Q10 [38], quercetin [39] N-acetylcysteine [40], monoclonal antibodies to cytokines (infliximab, tocilizumab [41]), and many other compounds and pharmaceuticals. The efficacy and success rates of pharmacological strategies are, as one can conclude, limited, with four possible reasons suggested: (1) vitamin antioxidants based upon in vitro studies may be inefficient scavengers in vivo; (2) detrimental ROS may be compartmentalized at sites within the cell that are inaccessible by the administered antioxidant; (3) antioxidants may possess toxicities that mask their advantageous effect; and (4) heterogeneity of the groups under study [42]. The use of nutritional approaches to treat and prevent diseases and to leveling unhealthy environment adverse effects comes from eating foods rich in antioxidants or food additives/drugs based on such foods [43] – помидоры; [44]). However, systematic or meta-analysis results suggest that eating antioxidants is more often harmless, but there is insufficient evidence on the positive effects of this for disease treatment or for the improvement of life quality [43,45,46], or the benefits are only revealed under a strictly defined set of measures or under a strictly defined disease stage with the observance of dosages [47-50]. Genetic approaches involve the regulation of the cell's own antioxidant system. The results obtained when discovering the targets for such regulation are reviewed below.»

  1. More data on cytokines that are involved in oxidative-related damages in human cancer cells ( interleukins and growth factors). 

We have added a short description of the "vicious cycle" of inflammation-oxidative stress before words about the role of oxidative stress in the pathogenesis of many diseases:

« Causality works both ways when it comes to the relationship between oxidative stress and inflammation. Through various signaling cascades, chronic oxidative stress can activate pro-inflammatory cytokines [9], and the inflammation itself, in turn, leads to the production of additional amounts of ROS [10]. The emerging vicious circle underlies many pathologies, including the initiation of carcinogenesis [9,11].»

Round 2

Reviewer 1 Report

The manuscript is improved and can be published.